# ORIENT: A Rank-Adaptive, Orthogonality-Preserving Neural Architecture

## Abstract

In this paper, we propose a novel neural network architecture ORIENT (ORthogonality-Intrinsic Elastic NeTwork), which is promising for various tasks including continual learning, light-weighting and so on. ORIENT is designed to intrinsically maintain strict orthogonality throughout training by parameterizing Givens rotations. Combined with a diagonal scaling component inspired by singular value decomposition, the architecture provides a compact and flexible factorization of weight matrices. It achieves high parallelism by arranging mutually non-interfering rotations at each layer, and supports rank-adaptive model construction by selective activation of Givens rotations. These properties enable ORIENT to efficiently represent low-rank approximations with a fixed structure. Hence, ORIENT is suitable for scenarios that require dynamic model reconfiguration. In the experimental evaluation, we show that ORIENT matches the performance of standard fully connected layers in the full-rank setting, while enabling smooth accuracy control through rank adjustment for light-weighting. In a continual learning task, progressive rank expansion allows effective capacity of growth while retaining previously learned knowledge.

Codes are available at `https://anonymous.4open.science/r/ORIENT-D017`.

## 1 Introduction

As deep learning models are deployed across a wide range of real-world applications, it is becoming increasingly important to flexibly adjust model architecture and capacity to suit practical deployment requirements Wang & Jia (2025). In particular, mobile and embedded environments require efficient model compression techniques to reduce computational cost while maintaining performance Saha & Xu (2025); Girija et al. (2025), whereas continual learning scenarios demand mechanisms to incorporate new tasks efficiently without compromising knowledge acquired from previous ones Wang et al. (2024).

Two properties *rank control* and *orthogonality* have been independently studied as key enablers of such flexibility, each demonstrating their respective effectiveness. Rank control allows dynamic scaling of model capacity. It enables pruning of unnecessary components during inference to reduce computational overhead while preserving accuracy, and it facilitates expansion of representational power when learning new tasks, by increasing rank as needed. Orthogonality structurally prevents interference between different tasks or representations. This makes it possible to add new tasks continually without degrading the performance on previously learned ones. Moreover, it improves the reusability of learned structures across different tasks or applications.

Despite the potential synergy of these two properties, existing methods tend to focus on one at the expense of the other, and no prior work has achieved a unified and rigorous integration of both. For instance, SVD-based training promotes low-rank structures but does not strictly enforce orthogonality Yang et al. (2020). Methods like Orthogonal Weight Modification (OWM) and Orthogonal Fine-Tuning (OFT) impose orthogonality only in the update directions, lacking mechanisms for flexible rank adjustment Zeng et al. (2019); Qiu et al. (2023). Massart & Abrol (2022) propose training methods that enforce strict orthogonality throughout optimization, though without supporting flexible rank adjustment. InRank dynamically increases matrix rank during training to improve efficiency, but it does not offer structural orthogonality after training Zhao et al. (2023).

In this context, what remains missing is a unified model structure that can simultaneously and explicitly support both strict orthogonality and dynamic rank control. In existing methods, orthogonality and rank adaptation are typically realized as algorithmic interventions or training-time constraints, rather than being embedded into the model architecture itself. This makes it difficult to adjust the model after training, reuse components across tasks, or scale the model's capacity on demand.

To address this gap, we propose ORIENT (ORthogonality-Intrinsic Elastic NeTwork), a novel linear transformation module that guarantees both strict orthogonality and elastic rank control at the architectural level. By leveraging Givens rotation-based parameterizations, the model is designed to maintain orthogonality consistently across all learning phases, including initial training, incremental learning, and continual adaptation. Moreover, ORIENT allows flexible rank control, enabling the model to adaptively size its representational capacity according to the demands of each learning phase. This architectural enforcement of orthogonality and rank control ensure that previously acquired knowledge is preserved while new knowledge can be safely incorporated. In addition, ORIENT achieves high parallelism since Givens rotations are arranged to avoid mutual interference at each layer. Figure 1 shows the example of ORIENT block. Figure 1(b) is a subnetwork of (a), designed such that the rank can be easily adjusted while minimizing the number of free parameters required for the change. Our contributions are summarized as follows:

- ORIENT is a novel neural network architecture that enforces strict orthogonality and enables elastic rank control. We formalize desirable properties for parameter efficiency and design ORIENT to satisfy them, enabling adjusting the model capacity in response to rank changes. Since ORIENT serves as replacement for a standard fully-connected layer, it integrates seamlessly into diverse architectures including Transformer.

- We present several showcases that demonstrate the effectiveness of ORIENT. In the case of rank decreasing, we demonstrate that ORIENT produces lightweight models while maintaining accuracy by reducing redundant parameters. In the case of continuous learning, ORIENT fully prevents catastrophic forgetting while saving the model's capacity.

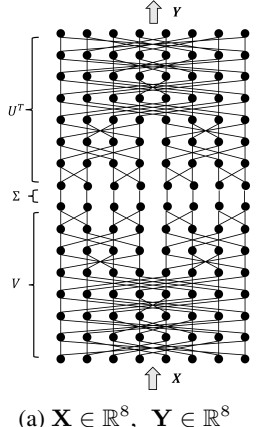

(a) $\mathbf{X} \in \mathbb{R}^8$, $\mathbf{Y} \in \mathbb{R}^8$

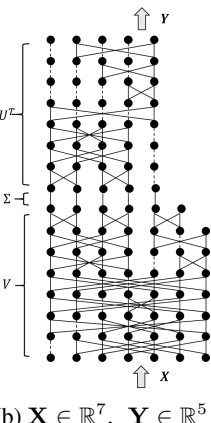

(b) $\mathbf{X} \in \mathbb{R}^7$, $\mathbf{Y} \in \mathbb{R}^5$

Figure 1: Examples of ORIENT block, whose input and output are $\mathbf{X}$ and $\mathbf{Y}$, respectively. Dashed vertical links in (b) represent dummy links, which have no-parameters.

## 2 PRELIMINARIES

Hereafter, we refer to Fully-Connected Layer as FCL. For simplicity of explanation, we consider FCL and ORIENT with $N = 2^n$ inputs and outputs as shown in Figure 1(a). Note that this assumption can be generalized to any $N$.

### 2.1 FACTORIZATION OF FULLY-CONNECTED LAYER

FCL is a standard architecture of a linear module. FCL is defined as follows. Let $\mathbf{X} \in \mathbb{R}^N$ and $\mathbf{Y} \in \mathbb{R}^N$ be input and output, respectively. And $\mathbf{Y}$ is defined as $\mathbf{Y} = \mathbf{X} \cdot \mathbf{W}^T$, where $\mathbf{W}$ is a weight

matrix. Let $\mathbf{W}$ be factorized by *Singular Vector Decomposition* (SVD) as $\mathbf{W} = \mathbf{U} \cdot \boldsymbol{\Sigma} \cdot \mathbf{V}^T$, where $\mathbf{U}^T \cdot \mathbf{U} = \mathbf{I}$ and $\mathbf{V}^T \cdot \mathbf{V} = \mathbf{I}$. In this case, the following formula holds: $\mathbf{Y} = \mathbf{X} \cdot (\mathbf{U} \cdot \boldsymbol{\Sigma} \cdot \mathbf{V}^T)^T = \mathbf{X} \cdot \mathbf{V} \cdot \boldsymbol{\Sigma} \cdot \mathbf{U}^T$ Then, $\mathbf{V}$ can be considered as an encoder, $\boldsymbol{\Sigma}$ as a scaler, and $\mathbf{U}^T$ as a decoder. $\mathbf{V}$ and $\mathbf{U}^T$ are unitary and orthogonal matrices.

Figure 2(a) and (b) are the network representations of the factorized FCL. As shown in Figure 2(b), rank reduction can be achieved by reducing the number of links in the $\boldsymbol{\Sigma}$ component.

When updating the weights of $\mathbf{V}$ and $\mathbf{U}^T$, it is necessary to continuously maintain the orthogonality. One of approaches to maintaining orthogonality is to use the Gram-Schmidt method or its derivatives Strang (2012). However, since this method is a type of greedy algorithm, the resulting solution is limited to being quasi-optimal.

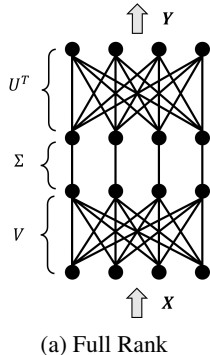 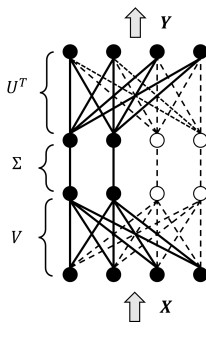

(a) Full Rank  (b) Low Rank

Figure 2: Examples of Factorized FCL blocks. The nodes of white circle and the dotted links in (b) are deleted for rank reduction.

## 2.2 GIVENS ROTATION NETWORK

Let $\mathbf{G}_{i,j}(\theta)$ be a Givens rotation matrix defined as Figure 3(a). Let $\mathbf{Z}^{(n)} = (z_1^{(n)}, \cdots, z_N^{(n)})$ be an input of $n$-th layer of a *Givens rotation network*, and $\mathbf{Z}^{(n+1)} = (z_1^{(n+1)}, \cdots, z_N^{(n+1)})$ be an output. Figure 3(b) shows a simple Givens rotation network, where $\mathbf{Z}^{(n+1)} = \mathbf{Z}^{(n)} \cdot \mathbf{G}_{i,j}(\theta)$. This kind of simple rotation is essentially parameterized by a single variable, $\theta$. When we focus only on the $i$-th and $j$-th values of $\mathbf{Z}^{(n)}$ and $\mathbf{Z}^{(n+1)}$, we can describe in short:

$$(z_i^{(n+1)}, z_j^{(n+1)}) = (z_i^{(n)}, z_j^{(n)}) \cdot \mathbf{P}(\theta_{i,j}), \quad \text{where } \mathbf{P}(\theta_{i,j}) = \begin{pmatrix} \cos\theta_{i,j} & -\sin\theta_{i,j} \\ \sin\theta_{i,j} & \cos\theta_{i,j} \end{pmatrix}.$$

Any unitary matrix can be represented by a sequence of products of Givens matrices Golub & Van Loan (2013). Since $\mathbf{U}$ and $\mathbf{V}$ are obtained by SVD, both matrices are unitary matrices. Therefore, $\mathbf{U}$ and $\mathbf{V}$ are represented as products of Givens matrices. As the number of types for N-dimensional rotation is $_NC_2$, both of $\mathbf{U}$ and $\mathbf{V}$ can be constructed with $_NC_2$-parameters, respectively. However, the ordering of the product is not unique.

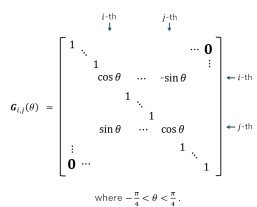 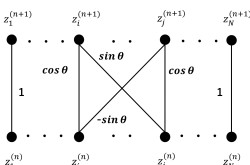

(b) Givens Rotation Network: $\mathbf{Z}^{(n+1)} = \mathbf{Z}^{(n)} \cdot \mathbf{G}_{i,j}(\theta)$. Label denotes a weight of the link.

(a) Givens Rotation Matrix

Figure 3: Givens rotation and its network representation.

## 2.3 ORDERING OF SINGULAR VALUES IN A SCALER

Let $\mathbf{Z}^{(n)} \in \mathbb{R}^N$, $\mathbf{Z}^{(n+1)} \in \mathbb{R}^N$ be input and output of a scaler component $\boldsymbol{\Sigma}$. And let $\sigma_1, \cdots, \sigma_N$ be parameters of $\boldsymbol{\Sigma}$. $\sigma_i$ is called as a singular value. In general, the ordering of singular values is not unique. For simplicity, we assume that the singular values are sorted in descending order and that $\sigma_i > \sigma_{i+1}$ holds. To satisfy this constraint, we introduce a sigmoid function $f$ with a parameter $\varsigma$. We define $\sigma_i$ as follows: $\quad \sigma_i = \begin{cases} A \cdot f(\varsigma_1) & (i = 1) \\ \sigma_{i-1} \cdot f(\varsigma_i) & (i > 1) \end{cases}$ , where $A$ is a constant. Because $0 < f(\varsigma) < 1$ satisfies, $\sigma_i > \sigma_{i+1}$ holds.

When we focus on the $i$-th value of input $\mathbf{Z}^{(n)}$ and output $\mathbf{Z}^{(n+1)}$, we can define: $\quad z_i^{(n+1)} = z_i^{(n)} \cdot \sigma_i = z_i^{(n)} \cdot A \cdot \prod_{k=1}^{i} f(\varsigma_k)$.

# 3 CHALLENGES

Orthogonal matrices constructed by Givens matrices products intrinsically maintain strict orthogonality. However, as mentioned above, the ordering of the Givens rotation's product is not unique. And determining the desirable ordering that achieves both high parallelism and elastic rank control is a non-trivial task.

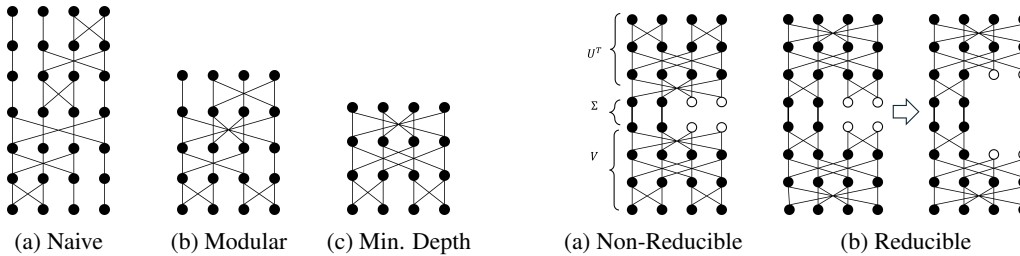

| (a) Naive | (b) Modular | (c) Min. Depth | (a) Non-Reducible | (b) Reducible |

Figure 4: Different ordering types of Givens rotation ($N = 4$).

Figure 5: Rank Control for the Givens Rotation Networks

## 3.1 HIGH PARALLELISM

Here we consider the order of Givens multiplication to achieve high parallelism.

Figure 4 shows three Givens Rotation Networks with the same input dimension, while differing in the production sequences of six Givens rotations. Figure 4 (a) is a naive sort order. In this subfigure, a single rotation is arranged on each layer level. Thus, it is guaranteed that no interference between rotations occurs at any level; however, the network depth reaches the maximum of $_NC_2$. In Figure 4 (b), multiple-rotations are packed in the same layer level. This packing is conducted based on modular operation, i.e. $(i + j) \bmod N$. Rotations with equal modulo, such as $\mathbf{G}_{1,2}(\cdot)$ and $\mathbf{G}_{3,4}(\cdot)$, are arranged on the same level. These rotations can be executed in parallel, because of no interferences. The last network (c) has $N$ layers, and achieves the minimum depth among Givens rotation networks with N-dimensional input. Between the layers, there are $N/2$ Givens rotation which can be applied at once. From the point of parallel processing, (c) is one of the best network with $N$-input. We call this type of the $N$-layers rotation network as *Minimum-Depth*. In Proposed Architecture section, we show an algorithm to construct a Minimum-Depth Structure for ORIENT.

## 3.2 ELASTIC RANK CONTROL

Here, we discuss the issues involved in elastic rank control using Givens rotation networks. In Figure 5, two types of network structures are shown. Each structure has the same input dimension and the same output dimension. Each one consists of two *Minimum-Depth* Givens rotation networks ($\mathbf{V}$ and $\mathbf{U}^T$) which are connected by $\boldsymbol{\Sigma}$ component. As in the case of Factorized FCL (Figure 2(b)), removing two links in $\boldsymbol{\Sigma}$ component in Figure 5(a) can reduce rank of the architecture. However, no-rotation is deleted. Thus, computational cost is not reduced in this case. On the other hand, in Figure 5(b), when removing two links in $\boldsymbol{\Sigma}$ component, we can see redundant nodes and links are

simultaneously deleted. This is an ideal rank reduction. In Proposed Architecture section, we show an algorithm to construct a *Reducible Structure* for ORIENT.

## 4 PROPOSED ARCHITECTURE

ORIENT consists of three consecutive components: (1) an encoder based on Givens rotation network $\mathbf{V}$, (2) a simple scaler $\mathbf{\Sigma}$, (3) a decoder based on Givens rotation network $\mathbf{U}^T$, as shown in Figure 1 (a). $\mathbf{V}$ and $\mathbf{U}^T$ in ORIENT are designed to be *minimum-depth* and *reducible*. In case of $N$-dimensional ORIENT, both of encoder and decoder have $N$-layers, and every layer of these components consists of $N$-nodes. Thus ORIENT block has $2N$-layers, and each layer consists of $N$-nodes.

In the following subsection, we introduce a two-dimensional, row-major order array called *Pair-Table* to construct desired encoder and decoder for given $N$.

### 4.1 PAIR-TABLE

We denote $\mathbf{PTBL}_N$ as a Pair-Table for $N$-dimensional Givens rotation network. $\mathbf{PTBL}_N[n][k]$ ($1 \leq n \leq N-1$, and $1 \leq k \leq N/2$) indicates a pair of two indices $(i,j)$ which specifies $\mathbf{G}_{i,j}$ between $n$-th layer and $(n+1)$-th layer.

Figure 6(a) is an example of Pair-Table. In this example, $\mathbf{PTBL}_8[1][1], \cdots, \mathbf{PTBL}_8[1][4]$ indicate $\mathbf{G}_{1,2}, \cdots, \mathbf{G}_{7,8}$ between first and second layer. And also $\mathbf{PTBL}_8[2][1], \cdots, \mathbf{PTBL}_8[2][4]$ indicate $\mathbf{G}_{1,3}, \cdots, \mathbf{G}_{6,8}$ between second and third layer. Figure 6(b) is Givens rotation networks corresponding to this table. $\mathbf{PTBL}_N$ is constructed recursively from $\mathbf{PTBL}_{N/2}$ as shown in Figure 6c.

By using this Pair-Table, we can construct $\mathbf{V}$ and $\mathbf{U}^T$ components in ORIENT. And note that rank control of the network based on this Pair-Table is easily realized as Figure 7.

### 4.2 FORWARD PROCESS

Full description of the forward algorithm is described in Appendix. To save the space, here we describe only the essence.

Let input of forward process in ORIENT be $\mathbf{Z}^{(1)} = (z_1^{(1)}, \cdots, z_N^{(1)})$.

The output $\mathbf{Z}^{(N)}$ of the encoder $\mathbf{V}$ is recursively defined as:
$$(z_i^{(n+1)}, z_j^{n+1}) = (z_i^{(n)}, z_j^{(n)})) \cdot \mathbf{P}_{i,j}(\theta_{i,j}) \text{ , where } (i,j) \leftarrow \mathbf{PTBL}_N[N-n][k] \text{ .}$$

The output $\mathbf{Z}^{(N+1)}$ of the scalar $\mathbf{\Sigma}$ is defined as:
$$z_i^{(N+1)} = z_i^{(N)} \cdot A \cdot \prod_{k=1}^{i} f(\varsigma_k).$$

And output $\mathbf{Z}^{(2N)}$ of the decoder $\mathbf{U}^T$ is also recursively defined as:
$$(z_i^{(n+1)}, z_j^{n+1}) = (z_i^{(n)}, z_j^{(n)})) \cdot \mathbf{P}_{i,j}(\phi_{i,j})^T, \text{ where } (i,j) \leftarrow \mathbf{PTBL}_N[n-N][k] \text{ .}$$

As the range of $\theta, \phi$ is $[-\frac{\pi}{4}, \frac{\pi}{4}]$, $\theta$ and $\phi$ are unsuitable for conventional optimizer such as SGD Thus we introduce $\tau = \tan 2\theta, \eta = \tan 2\phi$ as parameters of encoder and decoder respectively. Note that $-\infty < \tau, \eta < \infty$.

### 4.3 BACKWARD PROCESS

Full description of the backward algorithm is described in Appendix. Let input of backward process in ORIENT be $\frac{\partial L}{\partial \mathbf{Z}^{(2N+1)}} = (\frac{\partial L}{\partial z_1^{(2N+1)}}, \cdots, \frac{\partial L}{\partial z_N^{(2N+1)}})$.

The output of the decoder $\mathbf{U}^T$ is recursively defined as:
$$(\frac{\partial L}{\partial z_i^{(n)}}, \frac{\partial L}{\partial z_j^{(n)}}) = (\frac{\partial L}{\partial z_i^{(n+1)}}, \frac{\partial L}{\partial z_j^{(n+1)}}) \cdot \mathbf{P}_{i,j}(\phi_{i,j}),$$

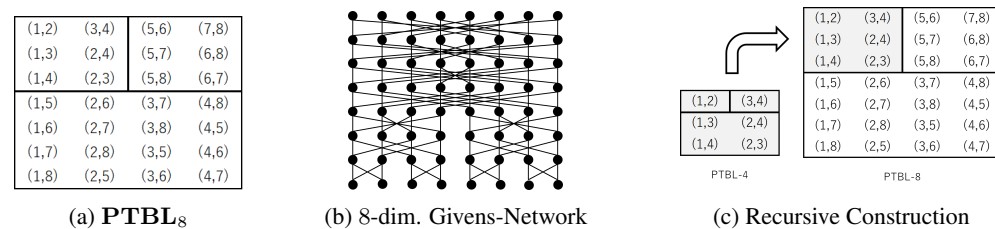

(a) $\mathbf{PTBL}_8$       (b) 8-dim. Givens-Network       (c) Recursive Construction

Figure 6: Construction of $N$-dimensional Givens Network based on Pair-Table.

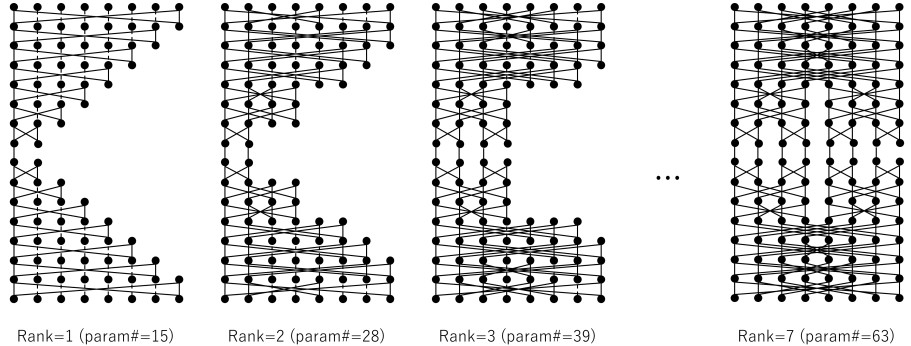

Rank=1 (param#=15)    Rank=2 (param#=28)    Rank=3 (param#=39)    Rank=7 (param#=63)

Figure 7: Elastic Rank Reduction Based on Sub-Networks in ORIENT

$$\frac{\partial L}{\partial \phi_{i,j}} = \left(\frac{\partial L}{\partial z_i^{(n+1)}}, \frac{\partial L}{\partial z_j^{(n+1)}}\right) \cdot \frac{\partial}{\partial \phi_{i,j}} \mathbf{P}_{i,j}(\phi_{i,j}) \cdot \begin{pmatrix} z_i^{(n)} \\ z_j^{(n)} \end{pmatrix} , \qquad \frac{\partial L}{\partial \eta_{i,j}} = \frac{\partial L}{\partial \phi_{i,j}} \cdot \frac{1}{2} \cdot \frac{1}{1+\eta^2} ,$$

where $(i,j) \leftarrow \mathbf{PTBL}_N[n-N][k]$ .

The output of the scalar $\mathbf{\Sigma}$ is defined as:

$$\frac{\partial L}{\partial z_i^{(N)}} = \frac{\partial L}{\partial z_i^{(N+1)}} \cdot \prod_{k=1}^{i} A \cdot f(\varsigma_k) ,$$

$$\left(\frac{\partial L}{\partial \varsigma_1}, \cdots, \frac{\partial L}{\partial \varsigma_N}\right) = \frac{\partial L}{\partial \mathbf{Z}^{(N+1)}} \odot \mathbf{Z}^{N+1} \cdot \begin{pmatrix} 1-f(\varsigma_1) & 0 & \cdots & 0 \\ 1-f(\varsigma_1) & 1-f(\varsigma_2) & \cdots & 0 \\ \vdots & \vdots & \ddots & \vdots \\ 1-f(\varsigma_1) & 1-f(\varsigma_2) & \cdots & 1-f(\varsigma_N) \end{pmatrix} .$$

And output of the encoder $\mathbf{V}$ is recursively defined as:

$$\left(\frac{\partial L}{\partial z_i^{(n)}}, \frac{\partial L}{\partial z_j^{(n)}}\right) = \left(\frac{\partial L}{\partial z_i^{(n+1)}}, \frac{\partial L}{\partial z_j^{(n+1)}}\right) \cdot \mathbf{P}_{i,j}(\theta_{i,j})^T ,$$

$$\frac{\partial L}{\partial \theta_{i,j}} = \left(\frac{\partial L}{\partial z_i^{(n+1)}}, \frac{\partial L}{\partial z_j^{(n+1)}}\right) \cdot \frac{\partial}{\partial \theta_{i,j}} \mathbf{P}_{i,j}(\theta_{i,j})^T \cdot \begin{pmatrix} z_i^{(n)} \\ z_j^{(n)} \end{pmatrix}, \qquad \frac{\partial L}{\partial \tau_{i,j}} = \frac{\partial L}{\partial \theta_{i,j}} \cdot \frac{1}{2} \cdot \frac{1}{1+\tau^2} ,$$

where $(i,j) \leftarrow \mathbf{PTBL}_N[N-n][k]$ .

## 5 EXPERIMENTAL RESULTS

We evaluate ORIENT on CIFAR-100 dataset Krizhevsky et al. (2009), which is widely used in image classification tasks including experiments in continuous learning settings. Figure 8 shows the model used in the following experiments. In this figure, $d_{\text{out}}$ takes different values depending on the task. Since ORIENT builds upon the standard FCL structure, we include FCL as a baseline for comparison. The two models share the same architecture, with the only difference being the use of FCL or ORIENT in the classifier. We modified the ResNet-50 architecture to suit the input size of CIFAR-100 and pre-trained it on Tiny ImageNet He et al. (2016); Le & Yang (2015) for use as a feature extractor. During training, the parameters of the feature extractor is frozen. To initialize ORIENT, we transform the weight matrix of FCL initialized using He initialization into the corresponding Givens rotation parameters He et al. (2015). We use Stochastic Gradient Descent as an optimizer Paszke et al. (2019). Details of settings for each scenario are provided at the beginning of the subsequent sections. The experimental settings are also available in Appendix.

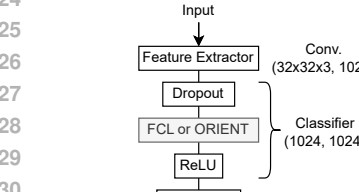

Figure 8: Model used in experiments

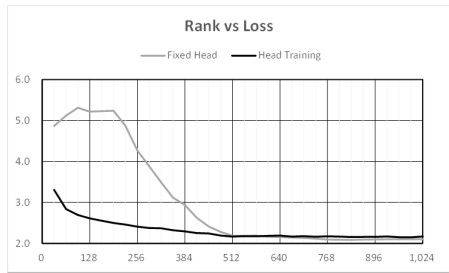

Figure 9: Validation loss curves. Both FCL and ORIENT achieved the same final accuracy of 46.7%.

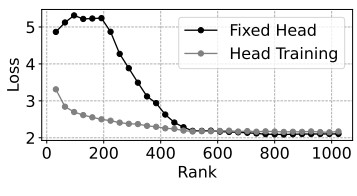

(a) Loss versus Rank

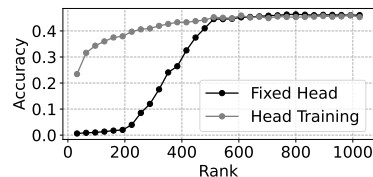

(b) Accuracy versus Rank

Figure 10: Rank-controlled inference with ORIENT.

## 5.1 FULL-RANK TRAINING

### 5.1.1 SETTING

We set $d_{\text{out}} = 100$ to classify 100 classes. Hyper parameters (initial learning rate, dropout rate, weight decay and so on) were tuned using Optuna Akiba et al. (2019) on the FCL model and subsequently shared with ORIENT. The tuning details are described in Appendix.

### 5.1.2 RESULTS

Figure 9 shows the validation loss curves for FCL and ORIENT. ORIENT achieves a similar loss to that of FCL throughout training, indicating that both the forward and backward computations in the proposed architecture section are functioning correctly. In fact, the final accuracy for both models was identical, reaching 46.7%. Although the hyperparameters were strictly shared between the two models, their learning dynamics did not completely match. This suggests that while the two parameterizations are theoretically capable of representing the same functions, the optimization paths taken by the models differ slightly due to their internal structure. Importantly, ORIENT imposes structural constraints such as strict orthogonality, yet it achieves comparable performance to the unconstrained FCL. This implies that its structured parameterization does not significantly limit its representational capacity. While the hyperparameters used for ORIENT were directly inherited from those optimized for FCL, the comparable performance suggests that ORIENT remains effective under such conditions. A more thorough hyperparameter search tailored specifically to ORIENT is left for future work.

## 5.2 RANK DECREASING

### 5.2.1 SETTING

In this experiment, we used full-ranked ORIENT model trained in the previous section. Hyperparameter settings are the same with the previous experiment. We evaluate the effect of reducing the rank of an ORIENT model at inference time. Two settings are compared: Fixed Head: The rank of the ORIENT model is reduced without any additional training. For each rank, the loss and accuracy are measured directly using the original classification head. Head Training: After reducing the rank of the ORIENT model, the backbone parameters are kept frozen, and only the classification head is retrained on the training set.

### 5.2.2 RESULTS

Figure 10 shows the effect of reducing the internal rank of ORIENT on validation loss and accuracy, under two settings: Fixed Head and Head Training. Even when the internal rank is reduced by approximately half, the performance degradation remains limited, indicating that the classifier head can still function effectively on representations with lower intrinsic dimensionality. This suggests that ORIENT's outputs retain sufficient discriminative information even when generated from low-rank internal structures. In lower-rank regimes, retraining the head leads to noticeable improvements in accuracy, indicating that the information available in the reduced-rank output can still be leveraged effectively if the head is adapted to the new distribution. These results highlight the robustness of ORIENT under rank constraints, and its potential for inference-time adaptability without full retraining.

### 5.3 CONTINUAL LEARNING (RANK INCREASING)

#### 5.3.1 SETTING

In this task-incremental learning setting, CIFAR-100 dataset is split into 10 disjoint tasks, each containing 10 classes. The common hyperparameters used in the previous experiments are the same. Each task has a separate classifier head with output dimension $d_{\text{out}} = 10$, and task identity is assumed known at inference time. We use the FACIL framework Masana et al. (2023) to conduct continual learning experiments. For each task, the model is first trained in a full-rank configuration. After training, we search for the lowest rank that satisfies predefined accuracy and loss margins on the validation set (We empathically decided the values of these parameters, which you can see in the code). The corresponding parameters are then frozen and reused as a fixed subspace for the next task. Subsequent tasks are also trained in full rank, followed by the same rank reduction and freezing process. As training progresses, frozen subspaces accumulate, and new tasks are learned only in the remaining unfrozen dimensions.

#### 5.3.2 RESULTS

Table 1 shows the task-wise and average accuracies for fine-tuning and ORIENT in a continual learning setting. While fine-tuning suffers from severe performance degradation on earlier tasks due to catastrophic forgetting (e.g., T1 drops from 81.0 to 75.4), ORIENT preserves performance across all tasks, maintaining 80.7% on T1 even after learning all 10 tasks. This supports the view that catastrophic forgetting is inherently avoided in ORIENT-based continual learning, as previously acquired parameters are efficiently preserved by its orthogonality. In addition, ORIENT outperforms fine-tuning in terms of average accuracy throughout training, achieving 76.7% at the final step compared to 74.8% by fine-tuning. Its performance remains stable in later tasks, demonstrating robust generalization and resilience to task interference. Table 2 illustrates the number of ranks selected in each task. The internal rank grows up to 928 during the initial tasks but does not continue to increase indefinitely. From Task 5 onward, the rank stabilizes around 768 and even decreases to 512 in the final tasks. This indicates that previously learned subspaces are reused effectively, and new tasks can be accommodated without over-expanding the model. In some cases, performance is maintained by updating only the task-specific head while keeping the shared representation unchanged. This supports the flexibility and efficiency of ORIENT in continual learning scenarios.

| Task | T1 | T2 | T3 | T4 | T5 | T6 | T7 | T8 | T9 | T10 | Avg. |
|---|---|---|---|---|---|---|---|---|---|---|---|
| Step 1 | 81.0 | - | - | - | - | - | - | - | - | - | **81.0** |
| Step 2 | 78.1 | 69.5 | - | - | - | - | - | - | - | - | 73.8 |
| Step 3 | 76.4 | 66.4 | 82.5 | - | - | - | - | - | - | - | 75.1 |
| Step 4 | 76.6 | 61.9 | 79.8 | 82.6 | - | - | - | - | - | - | 75.2 |
| Step 5 | 74.9 | 66.2 | 80.0 | 79.7 | 86.9 | - | - | - | - | - | 77.5 |
| Step 6 | 74.4 | 64.0 | 79.3 | 76.5 | 83.4 | 72.2 | - | - | - | - | 75.0 |
| Step 7 | 75.9 | 64.2 | 77.9 | 76.8 | 83.0 | 72.5 | 82.4 | - | - | - | 76.1 |
| Step 8 | 74.8 | 66.3 | 77.8 | 74.8 | 83.4 | 72.7 | 82.2 | 78.5 | - | - | 76.3 |
| Step 9 | 74.6 | 64.8 | 77.6 | 74.7 | 80.6 | 70.8 | 77.3 | 75.9 | 79.6 | - | 75.1 |
| Step 10 | 75.4 | 64.7 | 77.3 | 76.0 | 80.6 | 70.4 | 76.2 | 75.9 | 78.3 | 73.4 | 74.8 |

| Task | T1 | T2 | T3 | T4 | T5 | T6 | T7 | T8 | T9 | T10 | Avg. |
|---|---|---|---|---|---|---|---|---|---|---|---|
| Step 1 | 80.7 | - | - | - | - | - | - | - | - | - | 80.7 |
| Step 2 | 80.7 | 69.0 | - | - | - | - | - | - | - | - | **74.8** |
| Step 3 | 80.7 | 69.0 | 79.9 | - | - | - | - | - | - | - | **76.5** |
| Step 4 | 80.7 | 69.0 | 79.9 | 79.8 | - | - | - | - | - | - | **77.3** |
| Step 5 | 80.7 | 69.0 | 79.9 | 79.8 | 82.8 | - | - | - | - | - | **78.4** |
| Step 6 | 80.7 | 69.0 | 79.9 | 79.8 | 82.8 | 70.9 | - | - | - | - | **77.4** |
| Step 7 | 80.7 | 69.0 | 79.9 | 79.8 | 82.8 | 70.9 | 79.2 | - | - | - | **77.5** |
| Step 8 | 80.7 | 69.0 | 79.9 | 79.8 | 82.8 | 70.9 | 79.2 | 76.3 | - | - | **77.3** |
| Step 9 | 80.7 | 69.0 | 79.9 | 79.8 | 82.8 | 70.9 | 79.2 | 76.3 | 75.7 | - | **77.1** |
| Step 10 | 80.7 | 69.0 | 79.9 | 79.8 | 82.8 | 70.9 | 79.2 | 76.3 | 75.7 | 72.3 | **76.7** |

(a) Fine-tuning on $i$-th task.                    (b) Continual learning on ORIENT.

Table 1: Continual learning evaluation (accuracy).

| Task | T1 | T2 | T3 | T4 | T5 | T6 | T7 | T8 | T9 | T10 |
|------|-----|-----|-----|-----|-----|-----|-----|-----|-----|-----|
| Selected Rank | 512 | 672 | 768 | 928 | 704 | 768 | 768 | 768 | 512 | 512 |

Table 2: Number of sellected rank at the end of each step.

# 6 RELATED WORK

Orthogonality and rank control are key strategies for adjusting neural network capacity. These include enforcing orthogonal weight matrices and employing flexible model update techniques.

## 6.1 ARCHITECTURAL ORTHOGONALITY ENFORCEMENT

Architectural methods enforce weight orthogonality during training to improve stability and generalization. For example, orthonormal deep networks constrain weight matrices to be orthonormal, enhancing robust convergence Qin et al. (2024). Similarly, orthogonal constraints or regularizations on CNN filters promote feature diversity and mitigate gradient issues Rodríguez et al. (2017); Bansal et al. (2018); Wang et al. (2020). For RNN, Arjovsky et al. (2016); Wisdom et al. (2016); Jing et al. (2017); Massart & Abrol (2022) address vanishing gradients by using unitary or orthogonal weight matrices, enabling stable long-term memory. Li et al. (2019) proposes Orthogonal DNNs (OrthDNNs) with singular value bounding, yielding tighter generalization bounds. In model compression, SVD-training Yang et al. (2020) decomposes weights into orthogonal factors and singular values during training, adding orthogonality regularization to encourage a low-rank structure. These approaches improve initial training stability and network efficiency through enforced orthogonality, but, unlike ORIENT, they lack a mechanism to continuously maintain strict orthogonality during continual training or after model light-weighting.

## 6.2 FLEXIBLE MODEL UPDATES

Flexible update strategies mitigate forgetting especially in continual learning by restricting weight updates or expanding model capacity. Regularization-based methods such as Elastic Weight Consolidation (EWC) Kirkpatrick et al. (2017) and Learning without Forgetting (LwF) Li & Hoiem (2017) protect important weights or use knowledge distillation to retain prior knowledge. Projection-based approaches like Orthogonal Weight Modification (OWM) Zeng et al. (2019), Orthogonal Gradient Descent (OGD) Farajtabar et al. (2020), and Gradient Projection Memory (GPM) Saha et al. (2021) project new task gradients orthogonal to the subspace of previous tasks to prevent interference. Architectural expansion strategies such as Progressive Networks Rusu et al. (2016), Dynamically Expandable Networks (DEN) Yoon et al. (2018), and InRank Zhao et al. (2023) add new components dynamically to preserve knowledge while adapting to new tasks. For large pre-trained models, parameter-efficient fine-tuning such as LoRA Hu et al. (2022) adapts models with minimal weight changes to preserve prior performance; orthogonal fine-tuning (OFT) Qiu et al. (2023) and its structured variants like Butterfly OFT (BOFT) Liu et al. (2024) and quasi-Givens OFT (qGOFT) Ma et al. (2024) further constrain updates to orthogonal transformations. While these approaches mitigate forgetting, most require extra constraints or parameters and cannot fully eliminate forgetting unlike ORIENT.

# 7 CONCLUSION

We proposed ORIENT, a novel neural network architecture which can handle orthogonality and rank control. We employ Givens rotations to achieve these characteristics, and ensure high parallelism by arranging them in a manner that prevents mutual interference. With strict orthogonality and rank controllability, ORIENT is promising in many deep learning cases: continual learning, model compression, improving training efficiency, and so on. We conducted experiments in full-rank, decreasing-rank, increasing-rank settings, and demonstrated ORIENT functions as expected.

## ETHICS AND REPRODUCIBILITY STATEMENT

**Disclosure of AI assistance.**   We used a large language model to edit and polish the manuscript text. All research ideas, methods, and experiments were conducted solely by the authors.

**Data usage and privacy.**   All calibration and evaluation datasets used in this work are publicly available and contain no sensitive personal information. We used the data under their respective licenses, made no attempts at re-identification, and did not store or share model inputs and outputs beyond the scope of calibration and evaluation.

**Environmental impact.**   This study adds only a small incremental compute footprint: calibration consists of forward passes plus one small SVD per targeted sub-layer. We did not perform gradient-based fine-tuning in our experiments. At deployment time, structured pruning reduces parameter count and effective FLOPs, which can lower inference cost under comparable hardware and batching conditions.

**Fairness and safety.**   Structured pruning can alter performance unevenly across tasks, domains, or languages. We therefore evaluate on diverse benchmarks and report per-task metrics to surface potential regressions. No safety-critical deployment is claimed.

**Reproducibility.**   All experimental settings and tools are described in the main text, including the models and datasets and the calibration setup.

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

# A APPENDIX

## A.1 FORWARD PROPAGATION ALGORITHM

Algorithm of forward propagation of ORIENT is fully described in Algorithm 1. This algorithm assumes $N$-th dimensional ORIENT which has $N$-input and $N$-output. $\tau_{i,j}, \varsigma_k, \eta_{i,j}$ are parameters of an encoder $\mathbf{V}$ and a scaler $\boldsymbol{\Sigma}$, a decoder $\mathbf{U}^T$, respectively. The number of parameters of $\mathbf{V}, \boldsymbol{\Sigma}, \mathbf{U}^T$ are ${}_N C_2, N, {}_N C_2$, respectively. $\mathbf{PTBL}$ denotes a *Pair-Table*. $\mathbf{P}(\theta)$ is defined as follows:

$$\mathbf{P}(\theta) = \begin{pmatrix} \cos\theta & -\sin\theta \\ \sin\theta & \cos\theta \end{pmatrix},$$

where $-\frac{\pi}{4} \leq \theta \leq \frac{\pi}{4}$. $\tau$ and $\eta$ are used as parameters of Givens rotation matrix instead of $\theta$, $\phi$. $\tau$ and $\eta$ in this algorithm are defined as follows:

$$\tau = \frac{1}{2}\arctan\theta, \quad \eta = \frac{1}{2}\arctan\phi,$$

where $-\infty < \tau, \eta < \infty$. $f(\varsigma)$ is defined as follows:

$$f(\varsigma) = \frac{1}{1 + \exp(1 - \varsigma)}$$

.

In the last step of forward propagation in this algorithm, $\mathbf{Z}^{(1)}, \cdots, \mathbf{Z}^{(2N-1)}, \sigma_1, \cdots, \sigma_N$ are saved for use in backward process. $\mathbf{Z}^{(n)}$ is obtained recursively from $\mathbf{Z}^{(n+1)}$ in backward process, and $\sigma_i$ is also obtained from parameters $\varsigma_1, \cdots, \varsigma_N$ in backward process. Thus we can reduce the memory usage for these values at the expense of processing time in backward.

Note that loop (line 2 to line 6) and loop (line 14 to line 18) in this algorithm can be unrolled.

### A.1.1 EXECUTION EXAMPLE

For ease of understanding, a processing example in forward propagation with a small ORIENT block is shown in Figure 11. In this case, we assume $\mathbf{PTBL}$ as 3 x 2 array as follows:

$$\mathbf{PTBL} = \begin{bmatrix} [(1,2) & (3,4)] & [(1,3) & (2,4)] & [(1,4) & (2,3)] \end{bmatrix}.$$

When $n = 1, k = 1$ is specified in the loops of the encoder, $(z_1^{(2)}, z_4^{(2)})$ is obtained from $(z_1^{(1)}, z_4^{(1)})$. When $n = 4, i = 4$ is specified in the scaler, $z_4^{(5)}$ is obtained from $z_4^{(4)}$ and $\varsigma_4$. When $n = 6, k = 2$ is specified in the loops of the decoder, $(z_2^{(7)}, z_4^{(7)})$ is obtained from $(z_2^{(6)}, z_4^{(6)})$.

## A.2 BACKWARD PROPAGATION ALGORITHM

Algorithm of backward propagation of ORIENT is fully described in **??**. This algorithm assumes $N$-th dimensional ORIENT which has $N$-input and $N$-output. $\tau_{i,j}, \varsigma_k, \eta_{i,j}$ are parameters which are the same as in forward propagation algorithm. $\mathbf{PTBL}, \mathbf{P}(\theta)$ are also the same in forward propagation algorithm.

Note that loop (line 2 to line 8) and loop (line 13 to line 19) can be unrolled.

### A.2.1 EXECUTION EXAMPLE

For ease of understanding, a processing example in backward propagation with a small ORIENT block is shown in Figure 12. This example is corresponding to the forward execution example shown in Figure 11.

## A.3 ENVIRONMENT OF EXPERIMENTS AND THEIR HYPER PARAMETERS

Experiments are conducted under Rocky Linux 9.4 with A100-SXM4-80GB. As mentioned in *Experimental Results*, we use the *FACIL* framework Masana et al. (2023) in the experiments. Two hyper-parameters (i.e initial learning rate and dropout rate) were tuned with using *Optuna* Akiba et al. (2019). The tuning range was decided by the default manner of Optuna. These two hyper-parameters are commonly applied for constructing not only conventional models without ORIENT but the models with ORIENT. Other standard hyper-parameters (e.g. weight decay and so on) of FACIL are not tuned and determined empirically. These standard hyper-parameters are also applied for both conventional models and models with ORIENT. Two hyper-parameters are newly introduced for ORIENT. These new ones are not tuned and determined empirically. Used values of hyper-parameters are shown in Table 3, Table 4, and Table 5.

---

**Algorithm 1** Forward Propagation of ORIENT

**Require:**
$$\mathbf{Z}^{(1)} = (z_1^{(1)}, \cdots, z_N^{(1)}),$$
$$\tau_{1,2}, \cdots, \tau_{N-1,N},$$
$$\varsigma_1, \cdots, \varsigma_N,$$
$$\eta_{1,2}, \cdots, \eta_{N-1,N},$$
$$\mathbf{PTBL}[1 \ldots N-1][1 \ldots N/2]$$

**Ensure:**
$$\mathbf{Z}^{(2N)} = (z_1^{(2N)}, \cdots, z_N^{(2N)})$$

    { Forward in Encoder }
1: **for** $n = 1$ to $N - 1$ **do**
2:    **for** $k = 1$ to $N/2$ **do**
3:      $(i, j) \leftarrow \mathbf{PTBL}[N - n][k]$
4:      $\theta_{i,j} \leftarrow \frac{1}{2} \arctan \tau_{i,j}$
5:      $(z_i^{(n+1)}, z_j^{(n+1)}) \leftarrow (z_i^{(n)}, z_j^{(n)}) \cdot \mathbf{P}(\theta_{i,j})$
6:    **end for**
7: **end for**
    { Forward in Scaler }
8: $\sigma_1 \leftarrow \mathrm{A}(\text{const e.g. 1e+6}) \cdot f(\varsigma_1)$
9: **for** $i = 2$ to $N$ **do**
10:    $\sigma_i \leftarrow \sigma_{i-1} \cdot f(\varsigma_i)$
11: **end for**
12: $\mathbf{Z}^{(N+1)} = \mathbf{Z}^{(N)} \odot (\sigma_1, \cdots, \sigma_N)$
    { Forward in Decoder }
13: **for** $n = N + 1$ to $2N - 1$ **do**
14:    **for** $k = 1$ to $N/2$ **do**
15:      $(i, j) \leftarrow \mathbf{PTBL}[n - N][k]$
16:      $\phi_{i,j} \leftarrow \frac{1}{2} \arctan \eta_{i,j}$
17:      $(z_i^{(n+1)}, z_j^{(n+1)}) \leftarrow (z_i^{(n)}, z_j^{(n)}) \cdot \mathbf{P}(\phi_{i,j})^T$
18:    **end for**
19: **end for**
20: Save $\mathbf{Z}^{(1)}, \cdots, \mathbf{Z}^{(2N-1)}, \sigma_1, \cdots, \sigma_N$ to be used in the corresponding backward propagation.
    =0

---

| hyper-parameter | value |
|---|---|
| lr (initial learning rate) | 0.0263 |
| dropout (dropout ratio) | 0.4176 |

Table 3: Hyper-parameters tuned by Optuna. They are used commonly in the experiment.

| hyper-parameter | value |
|---|---|
| batch_size | 64 |
| clipping | 1.0 |
| lr_factor | 3.0 |
| lr_min | 1e-05 |
| lr_patience | 30 |
| momentum (for SGD optimizer) | 0.9 |
| nepochs (number of max epochs) | 300 |
| validation (ratio between validation num and train num) | 0.1 |
| weight_decay (for SGD optimizer) | 0.0 |
| num_exemplars (# of exemplars) | 0 |

Table 4: Standard hyper-parameters empirically determined. They are used commonly used in the experiment.

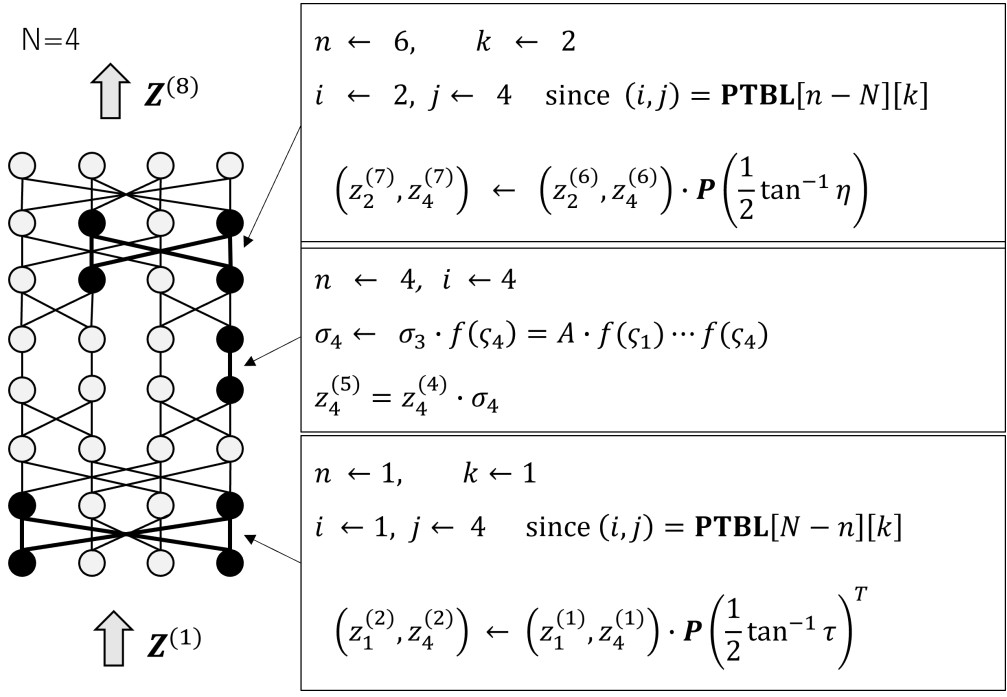

Figure 11: An execution example of Algorithm 1 with a small ORIENT block (4-dimensions).

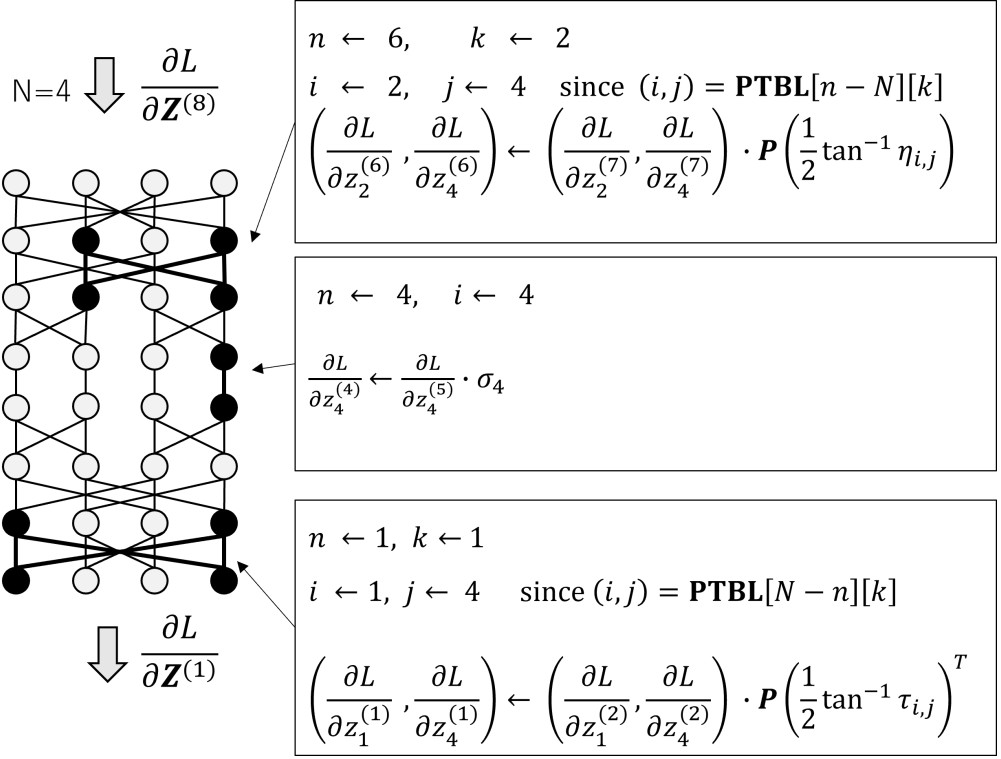

Figure 12: An execution example of Algorithm 2 with a small ORIENT block (4-dimensions).

| hyper-parameter | value |
|---|---|
| acc_margin (allowable amount of accuracy deterioration in reduced rank) | 0.004 |
| loss_margin (allowable amount of loss deterioration in reduced rank) | 0.006 |

Table 5: Newly introduced hyper-parameters for ORIENT. They are determined empirically.

