# OpenReview forum: "ORIENT: A Rank-Adaptive, Orthogonality-Preserving Neural Architecture"
_ICLR.cc/2026/Conference — Submitted to ICLR 2026_

### Official Review · Reviewer_ZtEL · 2025-10-29

**Soundness:** 1
**Presentation:** 2
**Contribution:** 1
**Rating:** 2
**Confidence:** 4

**Summary:**

This paper introduces a novel method to update low-rank linear layers while preserving orthogonality using Givens methods.

**Strengths:**

The method maybe beneficial in low-rank area of research by negating the necessity to reproject the U, V matrices in the SVD decomposition on to the Stiefel manifold, as is done in most works in this area.

**Weaknesses:**

-- The paper does not provide any result comparison with existing methods, for example:

    -- accuracy comparison with existing orthogonal weights methods ("Efficient Riemannian optimization on the Stiefel manifold via the Cayley transform", Li et al)
             -- with existing low-rank methods ("Pufferfish: communication-efficient models at no extra cost" Wang et al., " Low-rank lottery tickets: finding efficient low-rank neural networks via matrix differential equations" Schotthöfer et al)
             -- with existing low-rank AND orthogonal AND dynamical method ("Dynamical Low-Rank Compression of Neural Networks with Robustness under Adversarial Attacks" Schotthöfer et al)
    -- time comparison with regular FCL
             -- with existing low rank methods (see above)
    -- any other comparison that would be appropriate to show the advantages of the proposed method
-- The paper does not provide any motivation as to why their new method would be better than existing methods.

-- All papers mentioned above are also not addressed in the introduction, while being closely related to the area of this paper which signifies insufficient survey of literature.

-- Full description of backward pass is not provided, the reference on the line 684 is empty, thus the correctness of the method can not be fully verified.

**Questions:**

Please revise the experiment section to provide the comparison with the existing methods (mentioned in weaknesses) in accuracy, time or possibly memory and more importantly please add the motivation for why this method should be used compared to existing solutions.
It would also be beneficial to see other datasets.
Additionally I would be beneficial to show results of an architecture that consists only of these novel low-rank layers (even if just on a small dataset, such as any of the MNIST variations).

---

### Official Review · Reviewer_wLjL · 2025-10-29

**Soundness:** 2
**Presentation:** 1
**Contribution:** 2
**Rating:** 2
**Confidence:** 4

**Summary:**

The paper proposes ORIENT, an architecture for fully-connected layers that enforces strict orthogonality and enables adaptive rank control through a learned singular value parameterization. By parameterizing weight matrices via products of Givens rotations, ORIENT enforces strict orthogonality during all training phases without approximate projections. It integrates learnable singular value scaling with a recursive sigmoid parameterization (multiplicative ordering) to maintain ordered singular values, providing smooth rank adjustment. Although the premise is interesting, the paper could do a lot better with proper presentation, clear details, and extensive experiments.

**Strengths:**

1) The core idea of parameterizing layers with orthogonal Givens rotations linked to singular values is mathematically interesting.

2) The approach logically unifies orthogonality preservation with rank control.

3) The theoretical exposition of enforcing orthogonality via rotations is novel and could inspire further work. Also, for fully-connected networks, ORIENT demonstrated performance on par with traditional methods.

**Weaknesses:**

1) The paper is poorly presented and difficult to follow in key sections, including the main method and parameter update algorithms.

2) The evaluation is limited mostly to synthetic or standard fully connected layer tasks; lacking broader validation on diverse architectures and real-world benchmarks.

3) Computation/efficiency trade-offs from the Givens rotation parameterization versus standard layers are not addressed.

4) Important clarifications around singular value ordering and rank adaptation mechanisms are confusing or missing.

Minor Weaknesses: Some of the references are missing in the Appendix.

**Questions:**

1) Could the authors clarify the training stability and convergence behavior for any small fully-connected network?

2) Can you provide clearer algorithmic descriptions and pseudo-code for training steps?

3) Can the authors quantify the computational overhead relative to standard dense layers or other parameter reduction methods?

---

### Official Review · Reviewer_FRKD · 2025-11-01

**Soundness:** 3
**Presentation:** 3
**Contribution:** 3
**Rating:** 4
**Confidence:** 4

**Summary:**

This paper proposes a novel neural network architecture, ORthogonality-Intrinsic-Elastic-NeTwork (ORIENT), which decomposes the classifier head into a product of Givens matrices. This design enables the model to achieve both orthogonality and rank adaptation simultaneously, a combination not unified in prior work. The architecture is evaluated on standard image benchmark (CIFAR-100) under rank-decreasing conditions and in continual learning scenarios, particularly class-incremental learning on CIFAR-100.

**Strengths:**

- The paper is strong in clarity and logical organization. Because it introduces several unfamiliar architectures and design components, providing sufficient background is essential, and Sections 2 and 3 (*Preliminaries* and *Challenges*) do this very effectively. The figures (Figures 1–7) are also well-designed, making it easy for readers to follow the content visually.
- The proposed architecture, ORIENT, performs well on the standard CIFAR-100 benchmark even when the rank of the classifier head is significantly reduced. More importantly, it maintains strong performance without catastrophic forgetting in the continual learning setting, where both orthogonality and rank adaptivity are critical.

**Weaknesses:**

- As far as I understand, the proposed architecture ORIENT may be computationally inefficient and time-consuming. Since it decomposes an $N \times N$ matrix (a single layer) into $2N$ layers, using a high dimension such as $N = 1024$ would result in 2048 layers. Both forward and backward passes are required for all these layers, which could significantly increase computational cost. In addition, the total number of parameters would likely exceed that of a standard FCL. It would be helpful if the authors could report the computational cost and training time of ORIENT, along with the total number of parameters, and provide a comparison with a conventional FCL.
- The paper lacks a clear explanation of which components are trainable parameters. Please clarify explicitly in the main text which components of the architecture are learned during training.
- I would also like to raise concerns about the adaptivity of the proposed architecture. The current design appears to be applicable only to square matrices. In addition, it seems that ORIENT can be used only after the frozen feature extractor. How would this architecture be applied if the feature extractor itself needs to be modified or replaced?
- To validate ORIENT, I have several concerns regarding the experimental design:
    - The number of experiments is too limited. More diverse settings and datasets are needed. For example, for standard training, evaluations on datasets such as CIFAR-10 or SVHN would strengthen the analysis. For continual learning, experiments should include not only the class-incremental setting but also standard benchmarks like Permuted MNIST and Rotated MNIST.
    - In addition, similar to the setups in [1, 2], it would be useful to evaluate the *plasticity* of ORIENT beyond catastrophic forgetting. For instance, the authors could divide the same dataset (e.g., CIFAR-100) into i.i.d. chunks and incrementally train the model to observe how performance evolves as new chunks are introduced.
    - It appears that only a single random seed was used in the experiments. Since there is no mention of the number of runs or any reported standard deviation, the results seem to be based on a single trial. This raises concerns about statistical reliability. The authors should rerun each experiment with at least five different random seeds and report the mean and standard deviation. Without such analysis, the current results cannot be considered trustworthy.
- I am not entirely sure where the *orthogonality* property in ORIENT originates. While it is stated that $U$ and $V$ are constructed to be orthogonal by Givens matrices, their product $U \Sigma V$(which is ORIENT) is not necessarily orthogonal. To justify the claim of orthogonality, each orthogonal component should have a clear and distinct functional role, yet this is not clearly explained. This ambiguity likely arises from the lack of detail on how ORIENT is parameterized in the continual learning setting. Please provide a clearer explanation of how ORIENT is applied in this setting. In particular, specify which parameters are shared across tasks and which are task-specific, how orthogonality is maintained throughout training, and whether additional parameters are introduced for new tasks. Including a visual illustration, similar to the other figures in the paper, would also make this explanation easier to follow.
---
References

[1] On Warm-Starting Neural Network Training, In NeurIPS 2020.

[2] DASH: Warm-Starting Neural Network Training in Stationary Settings without Loss of Plasticity, In NeurIPS 2024.

**Questions:**

- Figure 9 appears to be incorrect. It should likely show the validation loss curve over training epochs; otherwise the statements in the Results section are hard to reconcile.
- The procedure for reducing rank is not described. Please clarify how rank is decreased and whether smaller singular components are truncated.
- What happens if the rank of FCL is also reduced? A direct comparison between reducing the ORIENT rank and reducing the FCL rank would be informative.
- I want to know the computation time and parameter counts for each setting.
- Please specify how the validation set is defined in the continual learning experiments. Is it split from the training set or taken from the test set?
- In Table 1, performance on previous tasks is retained, but the performance on new tasks appears lower than FCL. Can the authors analyze the cause and provide discussion or ablations?
- Is it possible to train ORIENT from the beginning with a reduced rank, instead of starting with the full rank and then decreasing it later? If so, how would the performance compare?
- What would happen if all layers, including the feature extractor, were trained jointly with ORIENT?
- How is the PTBL constructed? Is it predefined according to the dimension and kept fixed during training? Does that mean the method only applies to square matrices? Please clarify these details.
---
If the authors adequately address the issues raised in the Weaknesses and Questions sections, I would consider increasing my overall score.

---

### Meta-Review · Area_Chair_WBpv · 2025-12-04

**Summary:**

This paper has many reviewer concerns, including inefficient methodology, poor presentation, limited evaluation and a lack of comparisons to existing methods.

The authors have not responded to the reviewers, all of which propose reject. Given this, I also propose reject.

**Reviewer Concerns:**

There is no author response. All concerns remain outstanding.

**Reviewer Scores:**

I predict that no reviewers would have changed their scores in the absence of an author response.

---

### Decision · Program_Chairs · 2026-01-26

Reject